

# Weekly High Resolution Multispectral and Thermal UAS Mapping of an Alpine Catchment During Summer Snowmelt, Niwot Ridge, Colorado

Oliver Wigmore[1,2,3], Noah P. Molotch[2,4]

[1]Antarctic Research Centre, Victoria University of Wellington, Wellington, New Zealand
[2]Institute of Arctic and Alpine Research, University of Colorado Boulder, Boulder CO, USA
[3]Earth Lab, University of Colorado Boulder, Boulder CO, USA
[4]Jet Propulsion Laboratory, California Institute of Technology, Pasadena, CA, USA

*Correspondence to*: Oliver Wigmore (oliver.wigmore@vuw.ac.nz)

**Abstract**

Alpine ecosystems are experiencing rapid change, as a result of warming temperatures and changes in the quantity, timing and phase of precipitation. This in turn impacts patterns and processes of ecohydrologic connectivity, vegetation productivity, and

water provision to downstream regions. The fine scale heterogeneous nature of these environments makes them challenging areas to measure with traditional instrumentation, and spatiotemporally coarse satellite imagery. This paper describes the data collection, processing, accuracy assessment, and availability, of a series of ~weekly interval unmanned aerial system (UAS) surveys, flown over the Niwot Ridge Long Term Ecological Research site during the 2017 summer snowmelt season. Visible, near infrared, and thermal infrared imagery were collected. Our unique series of 5-25 cm multispectral and thermal

orthomosaics provide a unique snapshot of seasonal transitions in a high alpine catchment. Weekly radiometrically calibrated Normalized Difference Vegetation Index imagery can be used to track vegetation productivity at the pixel scale through time. Thermal imagery can be used to map the movement of snow melt across and within the near subsurface, as well as identify locations where groundwater is discharging to the surface. A 10 cm digital surface model and dense point cloud are also provided for topographic analysis of the snow free surface. Data summaries, citations, and DOIs are provided in the Data

Availability section. These datasets augment ongoing data collection within this heavily studied and important alpine site, and are made publicly available to facilitate wider use by the research community.

## 1 Introduction

The complex topography of mountain regions drives patterns of precipitation, wind, energy availability, and snow (Anderton et al., 2004; Beniston, 2006; Elder et al., 1991; Erickson et al., 2005; Fagre et al., 2003; Grünewald et al., 2010; Ives et al.,

1997; Trujillo et al., 2007) which create high degrees of spatiotemporal heterogeneity in geologic, environmental, ecologic and hydrologic processes (Bueno de Mesquita et al., 2018; Christensen, L. et al., 2008; Hermes et al., 2020; Litaor et al., 2008; Monson et al., 2002; Trujillo et al., 2012; Wieder et al., 2017; Zhang et al., 2018). A significant limitation to research in mountain regions is data availability at an appropriate spatial resolution to resolve key processes. Field measurements are often limited in quantity, quality and distribution. Furthermore, the high degree of spatiotemporal heterogeneity over short distances

makes the extrapolation and interpolation of point data to larger regions particularly problematic. Meanwhile, satellite data are often spatially and/or temporally too coarse to provide meaningful insight on fine scale processes. Rapid return sensors (e.g. Moderate Resolution Imaging Spectroradiometer - MODIS) have low spatial resolution, where a single 500m pixel may include considerable topographic variation and land surface features in heterogeneous mountain regions. Medium resolution sensors (e.g. Landsat) are often too coarse to differentiate surface hydrologic features (stream, ponds, springs), and





observations may be too far apart in time. High resolution satellites (e.g. Pleiades, Digital Globe/Maxar) and crewed aircraft can provide high spatial and temporal resolution imagery but at usually significant financial cost. In mountainous regions cloud cover is also a considerable challenge for air- and space-borne data collection. Unmanned aerial systems (UAS) can address many of these limitations by facilitating the collection of very high resolution imagery (centimetre scale) and digital elevation models (DEMs) on demand, without cloud cover and at low cost. However, their spatial extent and the radiometric quality of

the datasets is generally lower than for traditional earth observing systems. Observations from UAS can bridge the gap between point and pixel, affording the exploration of new ecohydrological questions in mountain ecosystems.

Here we present a novel UAS-borne elevation, multispectral, and thermal imagery dataset that was collected over a ~40 ha alpine subcatchment of the Niwot Ridge Long Term Ecological Research (NWT LTER) site in the Colorado Rockies, USA.

This study site has been an area of active alpine research and data collection for the last ~70 years, and is one of the most intensively studied alpine ecosystems in the world (Bjarke et al., 2021). The datasets were collected approximately weekly from late-June 2017 through mid-August 2017, comprising the period of summer snowmelt and vegetation growth. As such this dataset provides a unique snapshot of a critical ecohydrologic transition within a high alpine catchment. Full documentation for the datasets is provided here to facilitate wider use by other researchers and ease use and integration with

the array of existing collocated instrumentation within this important study site.

## 2 Study Site

The NWT LTER is located in the headwaters of the Boulder Creek watershed (Figure 1) and comprises a range of altitudinal environments and includes a number of different focussed study sites within it (Bjarke et al., 2021). Niwot Ridge is one of the most extensively studied alpine systems in the world and was a UNESCO Biosphere

Reserve from 1979-2017, in addition to currently being a United States Forest Service (USFS) experimental ecology reserve, and a National Ecological Observatory Network (NEON) study site. Here we focus on the upper reaches of the NWT LTER in the 'saddle catchment' (40°03'09.42" N, 105°35'29.62" W) which has an elevation range of ~3420-3620 m asl. The saddle catchment is a roughly 40ha alpine subcatchment of the Boulder Creek. The lower reaches (3420 – 3450 m asl) are densely forested primarily with Limber Pine (*Pinus flexilis*) and Lodgepole Pine (*Pinus contorta*). This gives way to Engelmann Spruce

(*Picea engelmannii*) and Subalpine Fir (*Abies lasiocarpa*) at higher elevation; here, krummholz are deformed by the strong winds, and function as points of localised snow accumulation on the leeward side. Alpine tundra extends above this treeline transition zone, and includes a variety of low shrubs, cushion plants, grasses, sedges, mosses and lichens (Walker et al., 2001). This region can be separated into five broad communities; dry meadow, moist meadow, wet meadow, rocky fellfields, and snow bed (May & Webber, 1982; Walker et al., 2001). The saddle catchment has become a recent focus of the NWT LTER's

research activities, coinciding with the installation of a dense network of soil moisture/temperature and precipitation sensors in 2017, and the establishment of long-term snow manipulation studies in 2018 (Bjarke et al., 2021). These activities complement ongoing observations of plant community composition at the plot level, and meteorologic data collection that have been made at the study site for the past decades. The saddle catchment lies within a NEON site (NIWO), established in 2015. NEON conducts annual airborne surveys (LiDAR and multispectral) of the site, and maintains meteorological, ecological

and hydrologic instrumentation nearby.



## 3 Data Collection

### 3.1 UAS Platforms

We used two different multirotor UAS platforms; a hexacopter and a quadcopter (Figure 2a). The multirotor platforms were custom designed for operation in high elevation mountain environments (4000-6000 m asl) (Wigmore et al., 2015, 2019; Wigmore & Mark, 2017). For this study we retuned the platforms to operate at 3500 m asl and to handle higher winds (~10-22 ms$^{-1}$) which requires faster speeds and motor response time. Both platforms are constructed of carbon fibre to reduce weight and improve rigidity. Total weight of the systems excluding sensor payloads is 2.8 kg (hexacopter) and 2.4 kg (quadcopter); at the study site elevation they are capable of around 15-20 minutes of flight (depending on wind speed) using a 4S 10,000 mAh lithium polymer battery. They are equipped with the Pixhawk V1 flight controller and are capable of fully autonomous flight, including waypoint navigation and survey grids. The UAS can be manually controlled with a 2.4 GHz remote control link; an RFD900+ 915 MHz telemetry downlink provides direct communication with the ground control station, from where survey progress and flight information (ground speed, altitude, attitude etc.) can be observed. Surveys are planned and managed through the Ardupilot Mission Planner ground control system, running on a Windows based field tablet.

The platforms were fitted with visible (RBG) (Canon S110), red/near infrared (NIR) (MAPIR Survey 2), and thermal infrared (TIR) (FLIR Vue Pro R 320) cameras, providing a total of six spectral bands (dual red bands). The red/NIR camera has a measured band wavelength interval from ~630-690 nm (peak at 660 nm) and ~810-900 nm (peak at 850 nm) (MAPIR, 2021), while for the RGB camera, the exact band wavelength is unknown. RGB and Red/NIR can be flown simultaneously (similar field of view) and are mounted on a vibration reducing plate (but no gimbal). The TIR camera is flown separately as it requires a three-axis gimbal for image stabilisation and closer spaced flight lines due to its much narrower field of view. All image capture is triggered by camera intervalometers (i.e. no connection to flight controller, which minimises potential points of failure). The Red/NIR and TIR cameras have this intervalometer feature built in. For the Canon S110, we installed the Canon hack development kit (CHDK) and loaded the KAP_UAV.lua script (CHDK, 2016), which allows automated control of numerous camera functions. The TIR camera is connected to the autopilot and geotags images with position and orientation information at the time of image capture, RGB and Red/NIR images can be geotagged after the fact by time stamp matching (but for this study were not).

### 3.2 Ground Control

In late May 2017 we permanently installed 15 visible ground control points (GCPs) (Figure 3). Targets were 30 cm sheets of fluorescent coroplast plastic, with a duct tape cross marking the centre (Figure 2b). We also installed and surveyed 9 thermal targets (Wigmore et al., 2019), however unfortunately these were not clearly visible in the imagery and were therefore not used as GCPs. Targets were distributed across snow free regions of the study area. To minimise propagation of errors in the SfM processing and limit model 'doming' (James & Robson, 2014; Tonkin & Midgley, 2016) we ensured targets were installed at the survey perimeter, and at the high and low points of the survey area. However, this was constrained by snow cover. To mitigate this an additional 22 natural feature GCPs (e.g. isolated flat boulders ~50-100 cm in diameter) were progressively surveyed as the snow melted to provide better GCP distribution over these areas (Figure 3). Furthermore, an additional 48 co-registration markers were identified in the 14 August orthomosaic; their position and elevation was extracted and these were used as extra markers for the earlier surveys when visible (Figure 3). Not all GCP's and co-registration markers were used for each date due to some being obscured by snow, not covered by the survey, poor visibility, and high error estimates. Details of which GCP's and markers were used for each survey are provided in their respective processing reports. A further 100 positions (Figure 3) were surveyed during the field season for other projects occurring at the site (Hermes et al., 2020), and are used as vertical check points for accuracy assessment of the 14 August DSM. These positions were not specifically surveyed for the



purposes of error assessment and thus do not have an ideal spatial distribution and may lie in areas of fine scale topographic heterogeneity (local high/low points). The 9 thermal GCPs were also used as vertical check points.

All GCPs and vertical check points were surveyed with a dual frequency L1/L2 Altus APS3 global navigation satellite system (GNSS) receiver, using a stop/go post processed kinematic (PPK) methodology (Figure 2b). Each position was occupied for 3

mins at a 1 Hz interval. Base station observations were collected from a permanent UNAVCO operated base station (station code: NWOT) (Larson, 2009) (Trimble NetR9 receiver with Trimble Zephyr Geodetic antenna) located near the tundra lab at NWT (max baseline >1.5 km), and five National Geodetic Survey (NGS) continuous reference stations (CORS) from the surrounding area (station codes: STBT, TMGO, P041, EC01, COFC; 1 Hz L1/L2 GPS/GLONASS observations were used in all cases. Rover positions were post processed against the base station network using Topcon Magnet Tools to an accuracy

threshold of 2 cm horizontally and 5 cm vertically, and mean standard deviation of point solutions were < 1cm (horizontal and vertical) for ground target GCPs and < 2cm (horizontal and vertical) for natural feature GCPs.

### 3.3 UAS Survey Flights

Seven survey flights over the saddle catchment were completed from 21 June to 14 August (Table 1). Due to high, gusty winds and unstable weather (afternoon thunder storms) we were unable to fly the exact same extent for each survey date. Flying

around the crest of the ridge was particularly difficult in high winds (>18 ms$^{-1}$) due to unstable eddies and downdrafts. We were unable to capture thermal imagery on 27 June and 5 July, due to rapidly deteriorating weather conditions and approaching thunderstorms. Flight lines and image capture intervals (every 3 seconds RGB and NIR, every 1 second TIR) were selected to produce >85 % front lap and >65 % side lap. For each date we captured ~400 RGB and ~400 NIR frames, and ~3000 TIR frames, ~20,000 frames in total, around 200GB of raw data. The UAS are capable of terrain following, either from Google

Earth base data (typically SRTM), or from custom DEMs. In this case we used a high resolution 1 m LiDAR DEM of the study site, which minimises the possibility of DEM terrain errors impacting automated return to launch procedures when flying downhill from the launch site. An above ground level (AGL) flight altitude of 120m was selected to maximise area coverage (and minimise flight time) and remain within the legal limits of our Certificate of Authorisation (COA# 2015-WSA-75-COA). This resulted in a ground resolution of approximately 4 cm for RGB and NIR photo frames. RGB images were stored as .jpg.

Red/NIR images were captured as raw 14 bit .tif (to enable later reflectance calibration with sufficient radiometric resolution). Thermal data were collected at roughly 25 cm ground resolution, in 14 bit raw .tif format (as opposed to FLIR's proprietary RJPG), which makes the data easier to work with in the processing and analysis stages.

### 4 Data Processing

### 4.1 SfM Workflow

All data were processed using the structure from motion (SfM) workflow as implemented in Agisoft Photoscan Pro V1.4. This commercial software has been widely used within the academic community and numerous resources discuss its workflow in more detail (Agisoft, 2016; Verhoeven, 2011; Wigmore & Mark, 2017, 2018). The core workflow used for this study is summarised below. Complete details for the processing settings for each date can be found in the respective processing reports.

- Image tie point generation, alignment, and sparse point cloud creation utilising band 1 of all images (RGB/NIR/TIR)
simultaneously. Key point limit set to 120,000, tie point limit at 30,000, with alignment accuracy set to highest value.
- Identification of surveyed GCP, and 14 August co-registration marker locations in each image they are visible. TIR images were not included in this step as RGB GCP's were not visible in TIR imagery.
- Optimisation of sparse cloud, i.e. forcing sparse cloud into real world coordinates of the GCPs.





- Thinning of sparse point cloud to remove outlier points based on parameters of reprojection error (<1), reconstruction uncertainty (<50), and projection accuracy (<10).
- Optimisation and sparse cloud thinning were done iteratively up to 3 times to bring error estimation under 1 pixel (3 cm pixel) and >0.01 m combined positional error.
- Dense cloud generation based on RGB imagery only, with high quality, and aggressive point cloud filtering settings.
- Triangular irregular network (TIN) mesh generated from RGB dense cloud.
- TIN mesh smoothed and orthomosaics produced for each imagery data set at both 5 cm (RGB, NIR) and 25 cm (NIR and TIR) pixel size.

After SfM processing the NIR geotif values were converted to surface reflectance in the red and near infrared bands using the MAPIR QGIS plugin in combination with images of the MAPIR surface reflectance calibration targets that were collected during the flight (Figure 2c). The near Lambertian calibration targets comprise three plates of varying reflectance: white (87 % reflectance), grey (51 % reflectance), and black (23 % reflectance), with known surface reflectance between 350 nm and 1100 nm. TIR imagery is converted to °C using the FLIR factory conversion factor, Equation 1.

Equation 1:  Ts °C = (TIR raw pixel value * 0.04 – 273.15).

Thermal images were processed simultaneously with RGB/NIR data for the respective survey date (workflow above). However, an additional step is included in the TIR data processing workflow to mitigate image vignetting. Thermal images captured with low cost uncooled microbolometers often suffer from image vignetting where temperatures measured at the edges of the frame are systematically lower than those closer to the image centre (Kelly et al., 2019). Built in non-uniformity correction (NUC) algorithms aim to mitigate these errors, but cooler measurements are often still returned from the image periphery. To mitigate this effect we applied a manually delineated ellipsoid vignette mask to each of the thermal images. This mask excludes data from the image edges from the final thermal orthomosaic.

To assess the horizontal positional error of the RGB imagery we identified 101 stable image features (large rocks) that were relatively evenly distributed across the 14 August 5 cm RGB orthomosaic. These are referred to as Hz check points. We then measured the horizontal offset between the 14 August image and each orthomosaic date for which the Hz check points were visible. Some Hz check points were not visible in the earlier dates due to snow cover. Offset statistics (relative to 14 August position) for each date and the entire series were then calculated. Vertical DSM error was assessed by comparing the 14 August DSM against the 109 surveyed vertical check points (above).

## 5 Results

### 5.1 Processing Results

Results of the processing accuracy, alignment errors, and GCP positioning accuracy are provided within the processing report pdf files for each survey date. Almost 19,000 individual images were collected and processed. The surveyed area ranged from 0.58 to 0.80 km$^2$ with a maximum ground sampling resolution of 4.11 to 4.22 cm (RGB). Reprojection error ranged from 0.783 to 1.1 pixels. Point cloud density for the 14 August survey was 146 pts/m$^2$, which is sufficient for DEM resolution to ~10 cm.

### 5.2 Multispectral and Thermal Mapping Data Overview

Figure 4 displays a full data stack of data collected on 11 July 2017, and includes RGB imagery, NDVI (from Red/NIR camera), and surface temperature from the TIR camera, draped on the 14 August DSM. Data available for each date is summarised in Table 2. Figure 5 displays RGB, TIR, and NDVI orthomosaics draped over the 14th August DSM for a select number of the survey dates. Surface processes of snowmelt and vegetation green-up are clearly visible in the images. The high resolution of the RGB data facilitates the delineation of different vegetation types and landcover classes. Changes in snow extent can be



readily delineated and quantified. The co-temporal TIR imagery provides an insight into the snowmelt fed surface and subsurface hydrologic pathways that are present, which are visible as cold streams across the landscape. The radiometrically

calibrated NDVI image series facilitates quantitative assessment of changes in vegetation health and productivity (NDVI pixel value) over the survey period. Figure 6 is a close-up view of different areas on the 21 June survey date showing TIR, RGB, and NDVI orthomosaics. Clearly visible in the thermal imagery are linear features of colder surface temperatures, these are suggestive of overland flow associated with snowmelt from the snow drifts (Figure 6a/b). Wind redistribution and deposition of snow is evident, with areas of snow accumulation visible on the leeward side of trees and in topographic low points (Figure

6c). Meanwhile the eastern side of the catchment is snow free for all survey dates (Figure 5). Site visits during the winter accumulation months indicate that this eastern edge is a wind-scour zone and is regularly cleared of snow during the winter months. Using the TIR imagery we can also hypothesise the existence of subsurface hydrologic pathways. For example, the wet meadows and surface ponds in Figure 6d are not visibly connected to surface meltwater channels, however, they retain water late in the summer and maintain cold temperatures throughout the summer, suggesting these systems are potentially fed

by ground water springs (Eschbach et al., 2017; Hoffmann et al., 2014; Lee et al., 2016; Wigmore et al., 2019).

**5.3 Positional Accuracy**

Horizontal accuracy for each survey date relative to the 14 August survey is shown in Figure 7, along with the combined (mean) offset for all dates. Mean offset for the Hz check points is 13.3 cm, with a median value of 9.4 cm, and an interquartile range of 6.2 to 15.7 cm. Figure 8 shows the spatial distribution of these errors. Here, circle size indicates magnitude of the

mean horizontal offset, while circle colour indicates the relative standard deviation of the offset errors. Horizontal error is higher at the periphery of the survey area, where image overlap is lower and camera geometry is worse. Within the area overlapped by all survey dates (yellow boundary line), mean horizontal offset error is mostly less than 15 cm, with lower relative standard deviation. Therefore, limiting analysis to the area within the overlap boundary and working at pixel resolutions greater than 20 cm should minimise the impact of horizontal offset (coregistration) errors on results.

Vertical accuracy for the 14 August DSM was assessed relative to GNSS surveyed positions. Figure 9 displays the ellipsoid height difference between the 14 August 10 cm DSM surface and the 109 surveyed GNSS vertical check points, as well as the 37 GCP locations (including targets and natural features). DSM elevation was subtracted from GNSS elevation, thus negative values indicate that the DSM elevation is greater than surveyed ground elevation. Mean GCP difference was -2.1 cm, with an interquartile range of -5.8 to 7.3 cm. As expected, this is very low as the SfM model is forced to fit these GCPs locations. For

the vertical check points the median difference is -8.9 cm, with an interquartile range of -29.0 to 9.9 cm. Data are visibly skewed negative, i.e. DSM higher than ground elevation, this is likely due to vegetation growth as has been reported previously (Li et al., 2020; Wigmore & Mark, 2018). Figure 10 shows the spatial distribution of the vertical errors. Three notable outliers (1.5 x interquartile range) are visible in the data (Figure 9) with errors around ~-100 cm; this is higher than expected. Inspection of Figure 10 shows that one of the outlier values is located at the NW edge of the survey area, and, is therefore likely a result

of doming in the DSM due to poor camera geometry (James & Robson, 2014). However, the other significant outliers are located around the middle of the survey area, where survey geometry is more robust. It is possible that in this case the errors are associated with the vertical check point data itself as these positions were not collected specifically for this application and may not have been collected over stable and relatively uniform areas suitable for assessing the accuracy of the DSM (Wigmore & Mark, 2017).



## 6 Data

### 6.1 Data Availability

All data are made available through the LTER Data Portal of the Environmental Data Initiative (EDI) (Table 3), and are made available under Creative Commons Attribution License, CC BY 4.0. Data are organised in six groups as follows: 1) 5 cm RGB orthomosaics (Wigmore, 2022b); 2) 5 cm and 25 cm R/NIR orthomosaics (Wigmore, 2022a); 3) 5 cm multispectral orthomosaics (Wigmore & Niwot Ridge LTER, 2021a); 4) 25 cm TIR orthomosaics (Wigmore & Niwot Ridge LTER, 2022b); 5) NDVI datasets (Wigmore & Niwot Ridge LTER, 2021b); 6) elevation datasets (14 August 2017) (Wigmore & Niwot Ridge LTER, 2022a). Full extent data includes 5 cm RGB, 5 cm NIR, 25 cm NIR, and, 25 cm TIR (where available) at the maximum survey coverage. Data along the survey periphery are likely to suffer from increased positional errors due to the reduced number of images (lower overlap) and relatively poor camera geometry over these regions; these areas should therefore be clipped prior to analytical use. Clipped multispectral data includes 5 cm RGB and R/NIR imagery stacked in a single multiband .tif file that has been clipped to the maximum extent covered by all survey dates, including a buffer to mitigate low quality data at the periphery. Spectral bands are as follows: B1 Blue (uncalibrated), B2 Green (uncalibrated), B3 Red (uncalibrated), B4 Red (calibrated surface reflectance centred at 660 nm), B5 NIR (calibrated surface reflectance centred at 850 nm). 25 cm TIR orthomosaics are all provided as a clipped version in which data have been clipped to the same boundary as above (though coverage may be lower for these TIR surveys), and, is provided as a 32 bit floating point raster with units of degrees Celsius. NDVI data compiles NDVI for each date into a single multiband stack where each band corresponds to a survey date in series; i.e. B1 NDVI 21 June, B2 NDVI 28 June, B3 NDVI 5 July, etc. Basic analytical layers derived from this stack are also provided, including the maximum/peak NDVI value from the NDVI stack, and the survey date on which this was measured, stored as a day of year integer value. Elevation data includes the 14th August DSM (as a .tif) and unclassified RGB coloured pointcloud (in .laz). The DSM is derived from the unclassified point cloud, and can thus be considered representative of vegetation canopy surface elevation. For much of the study site vegetation is very short (<5 cm) or absent (bare ground), and for these areas the DSM is equivalent to ground elevation; this is not the case for forested areas in the southern section of the study area. For all raster datasets individual metadata is included as an .xml file in the ESRI ArcMap format and point cloud data are accompanied by a .txt readme metadata file. Unclipped RGB 5cm datasets are also accompanied by the SfM processing report created by Agisoft Photoscan Pro software, which documents the processing parameters used and GCP error for each survey date.

### 6.2 Data Caveats and Considerations

Data are provided at both 5 cm (RGB, NIR) and 25 cm (NIR, TIR) spatial resolutions on aligned raster grids. Due to the lack of gimbal stabilisation (RGB and NIR) and high winds at the site the 5 cm imagery suffers from localised areas of image blur. Additionally, at this highest resolution the impact of potential offsets between the RGB and NIR bands and between dates is increased. Therefore, for analytical purposes it is recommended that the data are aggregated to a larger (>10 cm) spatial resolution. RGB data are uncalibrated and are likely unsuitable for analytical uses outside of mapping/classification. NIR data are calibrated to surface reflectance for each date and thus can be used to calculate reliable NDVI values that are stable across the data series. Thermal data have not been calibrated with more accurate near-surface Ts measurements, and thus can be assumed to be accurate to, at best, +/-5 % or 5 °C per the manufacturer's (FLIR) documentation. Relative Ts differences are much higher resolution, with 0.04 °C pixel sensitivity recorded by the sensor. Thermal vignetting has been minimised through the masking process (above). Trends in the thermal data can originate from both cooling/warming of the scene during the survey window (~1 hr), and from warming up of the thermal sensor. The latter was minimised by allowing the camera to warm up before image capture, however the former is not addressed. Consequently, some north-south striping in the thermal mosaics is visible for some dates (e.g. 21 July 2017), this is a result of brief temporal gaps in flying while changing batteries. This is a





common trade-off with small format thermal imaging over large areas. Because of the lack of reliability in absolute Ts recovery for these datasets it is recommended that users of the Ts data rely on them primarily for mapping thermal anomalies and relative differences, as opposed to deriving insight from absolute Ts values. File naming conventions, spatial resolution, and spectral bands are summarised in Table 2.

**7 Challenges of Collecting Multitemporal UAS Data in Mountainous Environments**

The complex topography and high degree of spatiotemporal variation present in mountain environments makes them an ideal environment for the application of high resolution UAS-based remote sensing campaigns. Our mapping campaign leveraged both the high spatial (centimetre) and temporal (weekly surveys) resolution benefits of UAS, while also capturing quantitative multispectral imagery over a relatively large area. There were a number of challenges to completing such a demanding survey
protocol in a relatively hard to reach mountainous environment. These challenges can be summarised as technical and environmental challenges.

**7.1 Technical Challenges**

Conversion of TIR to Ts for data collected from small microbolometer type thermal sensors is a significant technical challenge. These sensors are prone to vignetting (Kelly et al., 2019), and warm up rapidly which can bias measurements over time
(Dugdale et al., 2019). Uniform emissivity corrections do not account for variations in land cover (and thus emissivity) within the scene (Aubry-Wake et al., 2015), and despite the relatively short distance to target (~120m AGL), there are often sufficient atmospheric effects to introduce measurement errors (FLIR, 2018; Torres-Rua, 2017). Furthermore, as the final image is a mosaic created from images collected over a ~1hr window, warming or cooling of the scene can result in thermal trends within the orthomosaics. A number of solutions have been suggested to remedy these issues. Orthomosaics are frequently calibrated
to higher accuracy in-scene Ts measurements collected with non-contact infrared radiometers, temperature loggers, and higher quality thermal cameras (Kraaijenbrink et al., 2018; Torres-Rua, 2017; Wigmore et al., 2019). However, this bias correction doesn't account for changes in the scene temperature during the survey, or drift errors induced by changes in the temperature of the camera. Individual frames can be calibrated prior to mosaicking, e.g. by correction to widespread surface features with stable Ts (e.g. melting snow at 0 °C) (Pestana et al., 2019), however this is difficult to implement for large image collections,
especially if the thermally stable feature is not visible in all image frames. Orthomosaics and/or individual images can also be classified by landcover type, with different emissivity values applied as appropriate (Aubry-Wake et al., 2015). Perhaps, the most viable solution lies in technical innovations, for example the development of light weight heated external shutters, which may increase measurement accuracy by as much as 70 % (TeAx, 2019). For this study we were less concerned with the accurate measurement of absolute Ts, and more interested in mapping thermal anomalies and relative differences, for which
microbolometer sensors are perfectly capable (Dugdale et al., 2019; Harvey et al., 2016; Poirier et al., 2013; Wigmore et al., 2019). However, where accurate measurements of Ts are required these issues should be carefully considered and addressed in the planning stages.

For the collection of time series NDVI the ideal solution is to calibrate Red/NIR digital numbers to surface reflectance. This calibration requires the imaging of known reflectance targets at the same time as the UAS survey or the collection of incident
sunlight through a secondary instrument (e.g. Parrot Sequoia and Micasense Red Edge multispectral cameras). For direct comparison of Red and NIR surface reflectance the latter method is preferable as each image can be corrected individually, which accounts for variations in illumination (e.g. passing clouds) during the survey period. However, when comparing ratio indices, such as NDVI, changing illumination is less of an issue and therefore surface reflectance calibration based on reflectance targets is a suitable method. It is important, however, that camera settings, such as ISO, exposure length, and
aperture, remain constant during the survey, that sufficient radiometric depth is available (i.e. shooting in 12/16-bit RAW, as

opposed to 8-bit jpg), and that camera settings do not allow any part of the scenes pixel values to saturate at the high or low end (over or underexpose). Overexposure is of particular concern for snow covered areas where reflectance is high (Bühler et al., 2016; vander Jagt et al., 2015).

## 7.2 Environmental Challenges

Environmental challenges were primarily related to site-specific atmospheric conditions that are typical of mountainous environments, specifically, wind, and altitude. Wind speeds at ground level were often in excess of 10 ms$^{-1}$, with regular gusts at flight altitude (120 m) of over 20 ms$^{-1}$ which is at the upper limit of what many small UAS systems are designed to handle. A launch elevation of 3500 m asl is also sufficiently high to significantly reduce flight time due to lower air density. To deal with these two issues we overhauled our existing UAS platforms which were designed for higher elevation (>4500 m asl) but
lower wind speeds in Peru's Cordillera Blanca (Wigmore et al., 2019; Wigmore & Mark, 2017). Our system was able to operate reliably in wind gusts of up to ~22-24 ms$^{-1}$. Live observations from the ground station telemetry stream and flight logs showed periods of wind-induced pitch and roll compensation of up to 45° (the programmed limit). Designing UAS that are both robust and powerful enough to handle these forces is critical for reliable and repeatable mountain operations in sub-optimal conditions. The relatively lower elevation (3500 m asl as opposed to >4500 m asl in Peru), allowed us to add strength (and
consequently weight) to the system.  In this context, we used thicker gauge carbon fibre tubes and CNC aluminium (as opposed to plastic) joint connectors, motor mounts etc.; while also seeing a ~25-30 % increase in flight time, which increased from ~14 mins to ~18 mins on a 4S 10,000 mAh LiPO battery. Flight time is the critical limit on maximum survey-able area. Multirotor systems are particularly limited in this respect (compared to fixed wing platforms), however multirotor systems are generally better at handling wind gusts. Long flight time multirotor systems are increasingly available, however, these are usually either
very light weight and thus weaker or are powered by high-efficiency (low kilovolt, large propeller) power systems which have slower response times and often lower maximum flight speeds which limits their ability to deal with strong winds. Furthermore, these high efficiency systems are usually larger and heavier which limits their ability to be transported into the backcountry.

## 8 Conclusion

We presented the data acquisition and processing methodology for a unique high spatiotemporal resolution series of UAS
derived datasets that includes cm-scale resolution RGB, NIR and TIR imagery over a ~40 ha study area in the Colorado Rockies collected ~weekly over a summer snowmelt season. These data are spatially coincident with the recently heavily instrumented NWT LTER saddle catchment. Over 20,000 individual image frames were collected. These were processed using a SfM photogrammetric workflow and tied to absolute coordinates with a network of GNSS surveyed GCPs. Horizontal coregistration errors were assessed by comparing offset from the final (snow free) survey and had a mean error of less than 20
cm for all dates. Mean vertical accuracy of the DSM was 8.9 cm higher than GNSS surveyed position. A series of 5 cm (RGB, NIR) and 25 cm (NIR, TIR) orthomosaics are provided for each date, along with a stack of 25 cm NDVI orthomosaics and NDVI summary data (peak NDVI and peak NDVI day of year). Elevation data for the snow free (14 August 2017) survey include a DSM and high density point cloud. Together these datasets provide a unique snapshot of summer snowpack, snowmelt, surface temperature, and vegetation growth in a high alpine environment and facilitate the mapping and
quantification of environmental variables and ecohydrologic processes at unprecedented spatial resolution. Potential applications for these data are many, but include mapping of surface and subsurface hydrologic connectivity, tracking vegetation productivity and seasonal green-up, mapping ecosystem zonation, quantifying distribution and changes in snow cover, snow depth, and snowmelt. These data are made publicly available to facilitate broader use by the research community. These datasets leverage both the high spatial and temporal resolution of UAS data capture, while also collecting imagery across



multiple spectral bands. As such these data may facilitate advances in our understanding of spatially and temporally dynamic ecohydrologic process and connectivity within alpine environments.

**Author Contributions**

Both authors planned the objectives of the study. O. Wigmore collected, processed, and prepared all datasets, and prepared the draft manuscript. Both authors contributed to the revision and editing of the final manuscript.

**Competing Interests**

The authors declare that they have no conflict of interest.

**Acknowledgements**

We would like to thank members of the INSTAAR Mountain Hydrology Group, for assistance with field data collection (GNSS survey) and UAS operations. We would also like to thank member of the Niwot Ridge LTER programme and Mountain
Research Station for logistical support and assistance.

**Funding**

O. Wigmore was supported in part through funding from the University of Colorado 2016 Innovative Seed Grant (awarded to N. Molotch), the University of Colorado Earth Lab Grand Challenge, and the Niwot Ridge LTER program (NSF DEB – 1637686). Additional support in the form of GNSS equipment loan for the GNSS rover was provided by UNAVCO with
support from the National Science Foundation (NSF) and National Aeronautics and Space Administration (NASA) under NSF Cooperative Agreement [No. EAR-0735156]. Logistical support for this research was provided by the Niwot Ridge LTER program (NSF DEB – 1637686).



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



**Tables:**

| Survey Date | Imagery Collected |
|---|---|
| 21-Jun-17 | RGB, NIR, TIR |
| 27-Jun-17 | RGB, NIR |
| 5-Jul-17 | RGB, NIR |
| 11-Jul-17 | RGB, NIR, TIR |
| 18-Jul-17 | RGB, NIR, TIR |
| 25-Jul-17 | RGB, NIR, TIR |
| 14-Aug-17 | RGB, NIR, TIR |

**Table 1: List of all survey flight dates and imagery collected. TIR imagery were not collected on 27 June and 5 July due to deteriorating weather conditions and early thunderstorms.**

| File Name Extension | Type | Bands/Description | Spatial Resolution |
|---|---|---|---|
| *_RGB5cm_FullExtent.tif[1] | Unclipped RGB | B1-R, B2-G, B3-B | 5cm |
| *_NIR5cm_CALIBRATED_FullExtent.tif[2] | Unclipped NIR | B1-Rcal, B2-NIRcal | 5cm |
| *_NIR25cm_CALIBRATED_FullExtent.tif[2] | Unclipped NIR | B1-Rcal, B2-NIRcal | 25cm |
| *_TIR25cm_FullExtent.tif[3] | Unclipped TIR | B1-Ts in °C | 25cm |
| *_MultiB_RGBNIR.tif[4] | Clipped Multispectral | B1-B, B2-G, B3-R, B4-Rcal, B5-NIRcal | 5cm |
| *_TIR25cm_CropRGB.tif[3] | Clipped TIR | B1-Ts in °C | 25cm |
| NDVI25cm_Stack.tif[5] | NDVI | NDVI stacked by survey date order | 25cm |
| NDVI25cm_Max.tif[5] | NDVI | Maximum NDVI from all surveys | 25cm |
| NDVI25cm_PeakDOY.tif[5] | NDVI | DOY maximum NDVI measured | 25cm |
| 20170814_DSM10cm.tif[6] | DSM | Elevation (Ellipsoid Height m) | 10cm |
| 20170814_pointcloud.laz[6] | Point Cloud | Coloured with RGB (Ellipsoid Height m) | >100pts/m² |

[1](Wigmore, 2022b) ; [2](Wigmore, 2022a) ; [3](Wigmore & Niwot Ridge LTER, 2022b) ; [4](Wigmore & Niwot Ridge LTER, 2021a) ; [5](Wigmore & Niwot Ridge LTER, 2021b) ; [6](Wigmore & Niwot Ridge LTER, 2022a)

**Table 2: Summary of available datasets and filename extensions as provided on EDI portal.**

| Data Set | Citation | DOI |
|---|---|---|
| 5 cm RGB Orthomosaics | Wigmore, O. 2022. Uncalibrated RGB orthomosaic imagery from UAV campaign at Niwot Ridge, 2017. ver 1. Environmental Data Initiative. | https://doi.org/10.6073/pasta/073a5a67ddba08ba3a24fe85c5154da7 |
| 5 cm and 25 cm R/NIR Orthomosaics | Wigmore, O. 2022. Calibrated red/near infrared orthomosaic imagery from UAV campaign at Niwot Ridge, 2017. ver 1. Environmental Data Initiative. | https://doi.org/10.6073/pasta/dadd5c2e4a65c781c2371643f7ff9dc4 |
| 5 cm Multiband Multispectral Orthomosaics | Wigmore, O. and Niwot Ridge LTER. 2021. 5cm multispectral imagery from UAV campaign at Niwot Ridge, 2017 ver 1. Environmental Data Initiative. | https://doi.org/10.6073/pasta/a4f57c82ad274aa2640e0a79649290ca |
| 25 cm NDVI datasets | Wigmore, O. and Niwot Ridge LTER. 2021. 25cm NDVI data from UAV campaign at Niwot Ridge Saddle Catchment, 2017 ver 1. Environmental Data Initiative. | https://doi.org/10.6073/pasta/444a7923deebc4b660436e76ffa3130c |
| 25 cm Thermal Infrared Orthomosaics | Wigmore, O. and Niwot Ridge LTER. 2022. Surface temperature mapped from thermal infrared survey from UAV campaign at Niwot Ridge, 2017. ver 2. Environmental Data Initiative. | https://doi.org/10.6073/pasta/70518d55a8d6ec95f04f2d8a0920b7b8 |
| Elevation datasets (14 August 2017) | Wigmore, O. and Niwot Ridge LTER. 2022. Photogrammetric point cloud and DSM from UAV campaign at Niwot Ridge, 2017. ver 2. Environmental Data Initiative. | https://doi.org/10.6073/pasta/1289b3b41a46284d2a1c42f1b08b3807 |

**Table 3: Data availability, citations and DOIs.**



**Figures:**

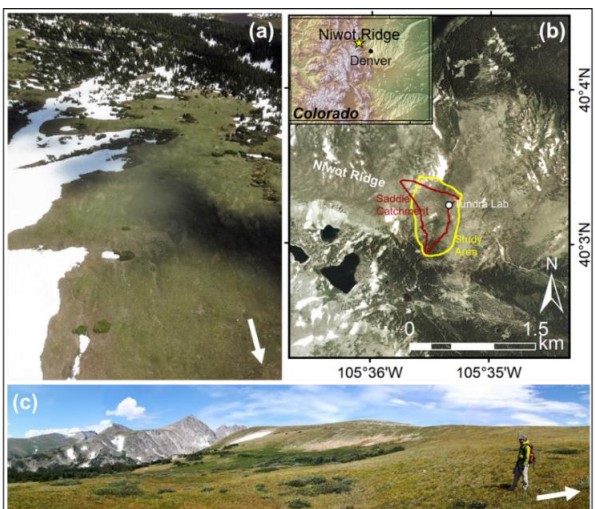

**Figure 1: Study site at Niwot Ridge (white arrows on photos indicate north). Base imagery in 1b) sourced from public access USA National Agricultural Imagery Program (NAIP) 2005.**

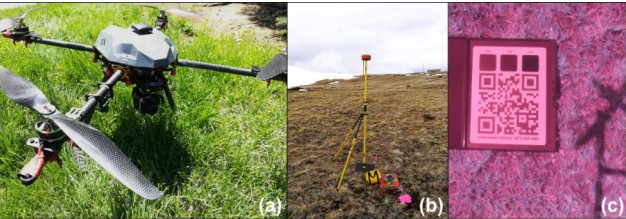

   **Figure 2: Quadcopter UAS fitted with thermal camera (a), installation and survey of GCPs (b), NIR/Red image of surface reflectance**
**calibration plate.**

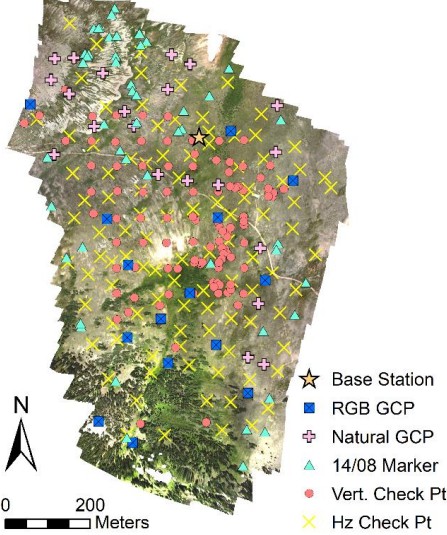

**Figure 3: Location of local base station, all GCPs, and check points used in SfM processing and error assessment.**



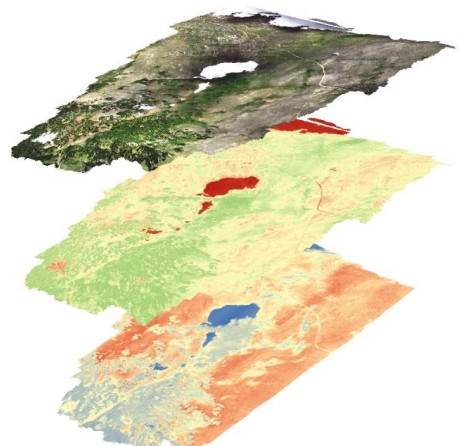

**Figure 4: Full data stack collected on 11 July 2017. Showing: RGB (top), NDVI (middle), and Ts (bottom), draped over 14 August 2017 DSM.**


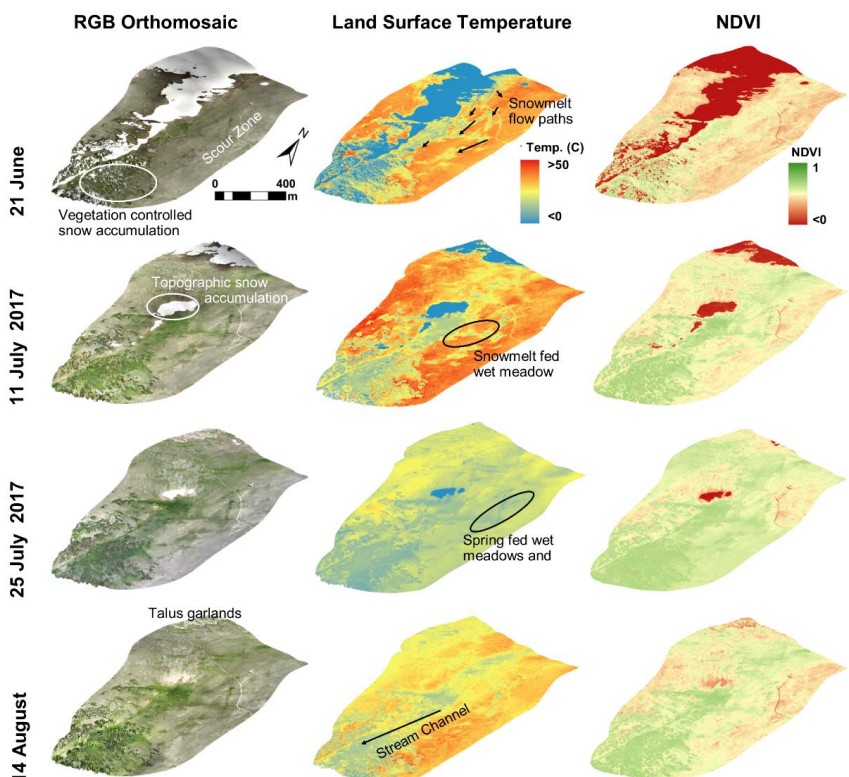

**Figure 5: 3D views of selected RGB orthomosaics (5cm), Ts maps (from TIR) (25cm), and NDVI (25cm) draped on 14 August 10cm DSM.**

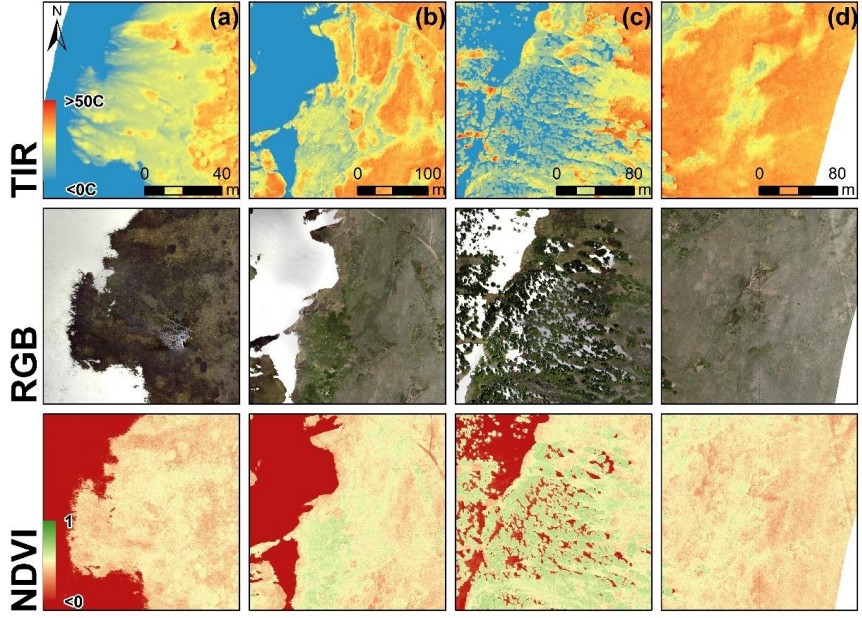


**Figure 6: Close up of TIR (top row) (25cm), RGB (middle row) (5cm), and NDVI (bottom row) (5cm) image pairs for 21 June 2017. (a) snow melt pathways through wet meadow; (b) snow melt feeding wet meadow; (c) snow accumulation in the forest zone; (d) surface ponds fed by subsurface hydrologic pathways.**

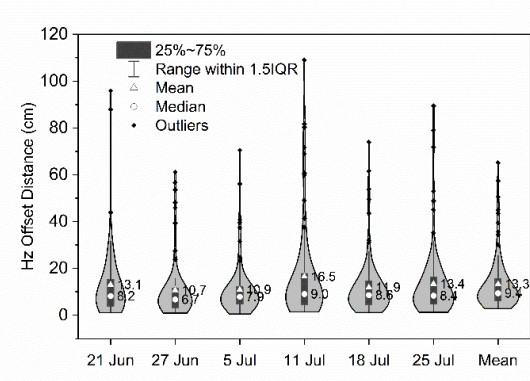

**Figure 7: Horizontal offset at check point locations for each survey dates RGB 5cm orthomosaic, relative to 14 August position. Mean horizontal offset is shown on right violin plot. Mean and median values labelled accordingly. Curve represents kernel smoothed frequency distribution of offsets.**



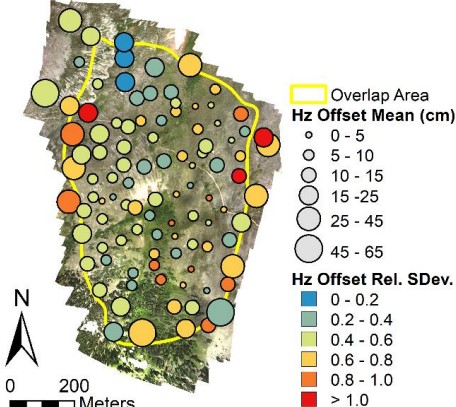

**Figure 8: Spatial distribution of horizontal offset errors, compared to 14 August position.**

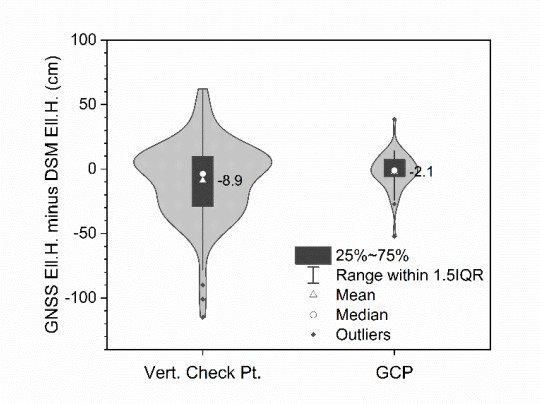

**Figure 9: Difference between GNSS ellipsoid height and 14 August DSM ellipsoid height for 109 check points and 37 GCPs. Negative values indicate DSM elevation is greater than GNSS surveyed elevation. Mean difference (cm) is shown on plot. Curve represents kernel smoothed frequency distribution of vertical differences.**


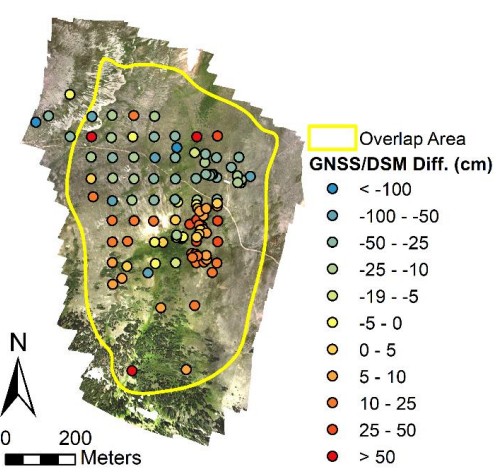

**Figure 10: Spatial distribution of vertical errors for 14 August DSM (GNSS ellipsoid height at surveyed check points minus DSM ellipsoid height). Negative values indicate DSM elevation is greater than GNSS surveyed elevation.**