# Peer review of "Weekly High Resolution Multispectral and Thermal UAS Mapping of an Alpine Catchment During Summer Snowmelt, Niwot Ridge, Colorado"

_Earth System Science Data, 2022_

## Author Response (AR1)

We would like to thank the reviewers, Paul Schattan and Marc Adams for their reviews and the Handling Editor James Thornton for his suggestions and input regarding the manuscript. We have compiled reviewer and editor comments below in black, our response to these comments is presented in blue italics, with changes specified in red italics. All changes can be seen in the tracked changes version of the revised manuscript.

**RC1**

This article introduces a multi-temporal and multi-spectral dataset of a mountain basin in the Western United States. Together with its high spatial resolution this data provides a unique basis not only for investigating local processes, but e.g. also for developing and/or validating new approaches using space-borne platforms. The processing steps including data accuracy and potential limitations are well documented.

*We thank the reviewer Paul Schattan for reading and commenting on the manuscript. We are glad that the uniqueness of the dataset and its potential use were recognised, and the comments re documentation of our work-flow. For our revised manuscript we will complete a thorough edit for clarity and typos to identify any issues that may have been missed.*

*We have completed a close proofread and edit of the manuscript for clarity, typos and formatting errors.*

**RC2**

In this contribution a series of seven datasets is presented, collected almost weekly during the 2017 melt season in the Colorado Rockies at around 3,500 m ASL. RGB, NIR and TIR imagery was collected over a 40 ha study site using a custom-built multi-rotor system. All imagery was processed using SfM-photogrammetry to generate corresponding orthorectified mosaics. These are complemented by an NDVI mosaic generated from radiometrically calibrated imagery. Extensive and robust ground control and error handling ensured cm-level georeferencing. The site, data collection and processing, as well as accuracy and challenges are very well described and documented. Overall, the manuscript is clearly structured, succinct and very well written - a joy to read! Figures and Tables are well presented and organised, captions are clearly written and comprehensible.

*We would like to thank the reviewer Marc Adams for his comments and revision of the manuscript. We have included responses to the specific comments raised in blue italics below each bullet point in the attached pdf. Furthermore, we will complete a thorough edit for clarity and typos to identify any issues that may have been missed.*

*We have completed a close proofread and edit of the manuscript for clarity, typos and formatting errors. We have addressed all specific comments raised by the reviewer and commented in red italics below.*

For specific and technical comments, please refer to the attached PDF.

**Specific comments**

From what I saw, only a single SfM-DSM was published from the snow-off flight? Since RGB-imagery from all dates was processed and ground control is available for all dates, it might be interesting to add DSMs and/or DPCs from the snow-on dates too. This could give potential users of the dataset the possibility to investigate the evolution of snow depth as well as snow melt patterns during the studied time frame.

*Agreed this is a useful dataset. The snow depth data are currently being QA QCd against additional snow depth validation data and are being released as part of a different publication addressing specific science questions around snow depth.*

**Technical comments (non-exhaustive)**

• Line 17: Consider using the term 'uncrewed', rather than 'unmanned' (corresponding to 'crewed aircraft' mentioned in line 40) – here and throughout the manuscript

*Agreed, we will make this change*

*Done*

• Line 19: Our unique […]

*We're unsure what this refers to?*

*Changed to "this unique"*

• Line 19: […] 5-25 cm […] for consistency here and throughout the manuscript i) consider switching to SI-units i.e. only use meters rather than switching back and forth between cm, m and km; ii) I advise using a hard space between numbers and units

*We will maintain SI units where suitable. However when working with this high resolution cm scale imagery we feel it is often easier to use appropriate non SI units e.g. cm. We will make changes to the manuscript where this may cause confusion.*

*Agreed re hard spaces, and this is the preferred format for ESSD. Most of are already formatted this way, however some may have been missed. We will complete a careful proof read of the manuscript to catch any that are incorrectly formatted.*

*Done – track changes throughout*

• Line 21: […] Normalized Difference Vegetation Index imagery […] → maps? calculations? mosaics? I'd reserve the term 'imagery' for the actual imagery, rather than processed products

*Agreed, we will make this change*

*Done changed to "maps"*

• Line 21: […] vegetation productivity at […] → maybe better to use 'vitality' here

*Agreed, we will make an appropriate change, e.g. health/vitality*

*Done changed to "health"*

• Line 23: A 10 cm High-resolution digital surface model [GSD 0.1 m] and dense point cloud

[add point density] are […]

*Agreed, we will make this change*

*Done - added point cloud resolution "146points/m2"*

• Lines 28ff: 16 references in a single sentence strike me as being a touch too many – consider

reducing to the most relevant ones.

*Agreed, we will make this change*

*Number of citations here reduced*

• Lines 32ff: …and conversely maybe add some references to the rest of the paragraph. Although of course this reasoning follows common sense for any reader with a background in remote sensing, a few well-placed references could nicely underpin your arguments.

*Agreed, we will make this change*

• Line 56: Here or better in the Figure 1 caption, you might want to add a brief description of a), b) and c), to the effect of 'overview / terrestrial / aerial image on the study site…' - for completeness' sake

*Agreed, we will make this change*

*Done, added to figure caption – tracked changes*

*"Figure 1: Study site at Niwot Ridge (white arrows on photos indicate north). a) oblique aerial image of the study site looking down slope (south) from the northern edge, b) study site location and relevant boundaries, c) terrestrial view of the study site from the eastern edge of the study site looking west. Base imagery in 1b) sourced from public access USA National Agricultural Imagery Program (NAIP) 2005."*

• Line 135: […] a high resolution 1 m [..]

*We're unsure what this refers to?*

• Line 135ff: I would have thought the main reason was to sustain a defined ground sampling distance over the whole study site. Not sure I get the point raised here about DEM errors (only) impacting automated RTL.

*In this case we are talking more about operational/navigational challenges, and will edit to make this clearer. However, we agree with the point re flight altitude above ground level and consistent ground sampling resolution. We will edit this appropriately.*

*Done tracked changes:*

*"In this case we used a 1 m LiDAR DEM of the study site, which helps to maintain consistent ground sampling distance during the survey and minimises the possibility of DEM terrain errors impacting automated return to launch procedures when flying downhill from the launch site"*

• Line 139: […] in a ground resolution of […] Could be substituted by ground sampling distance (GSD)

*Agreed, we may make this change in our final proof read.*

*Edited accordingly see tracked changes*

• Line 188: […] a full data stack of data collected […]

*We're unsure what this refers to? However grammatically it appears awkward, we will edit as appropriate.*

*Edited see tracked changes:*

*"Figure 4 displays a full stack of data collected on 11 July 2017, and incl"*

Editor Comment:

Given the strong emphasis on snow, some readers may appreciate a brief explanation of why NDSI maps were not also computed and provided (presumably because the requisite bands were not all available). I would ask to to consider this when you make any eventual revisions.

*NDSI requires visible and short wave infrared bands, typically calculated as (Green – SWIR) / (Green + SWIR). We did not collect any SWIR imagery so calculating NDSI was not possible.*

*We have not made any changes/additions regarding this.*